# VARIATIONAL AUTO-ENCODER ARCHITECTURES THAT EXCEL AT CAUSAL INFERENCE

## ABSTRACT

This paper provides a generative approach for causal inference using data from observational studies. Inspired by the work of Kingma et al. (2014), we propose a sequence of three architectures (namely Series, Parallel, and Hybrid) that each incorporate their M1 and M2 models as building blocks. Each architecture is an improvement over the previous one in terms of estimating causal effect, culminating in the Hybrid model. The Hybrid model is designed to encourage decomposing the underlying factors of any observational dataset; this in turn, helps to accurately estimate all treatment outcomes. Our empirical results demonstrate the superiority of all three proposed architectures compared to both state-of-the-art discriminative as well as other generative approaches in the literature.

## 1 INTRODUCTION

As one of the main tasks in studying causality (Peters et al., 2017; Guo et al., 2018), the goal of Causal Inference is to figure out **how much** the value of a certain variable would change (*i.e.*, the *effect*) had another certain variable (*i.e.*, the cause) changed its value. A prominent example is the counterfactual question (Rubin, 1974; Pearl, 2009) "Would this patient have *lived longer* [and by **how much**], had she received an alternative treatment?". Such question is often asked in the context of precision medicine, which attempts to identify which medical procedure $t \in \mathcal{T}$ will benefit a certain patient $x$ the most, in terms of the treatment outcome $y \in \mathbb{R}$ (*e.g.*, survival time).

A fundamental problem in causal inference is the unobservablity of the counterfactual outcomes (Holland, 1986). That is, for each subject $i$, any real-world dataset can only contain the outcome of the administered treatment (aka the *observed* outcome: $y_i$), but not the outcome(s) of the alternative treatment(s) (aka the *counterfactual* outcome(s)) — *i.e.*, $y_i^t$ for $t \in \mathcal{T} \setminus \{t_i\}$. In other words, the causal effect is never observed (*i.e.*, missing in any training data) and cannot be used to train predictive models, nor can it be used to evaluated a proposed model. This makes estimating causal effects a more difficult problem than that of generalization in the supervised learning paradigm.

In general, we can categorize most machine learning algorithms into two general approaches, which differ in how the input features $x$ and their target values $y$ are modeled (Ng & Jordan, 2002):

**Discriminative methods** focus solely on modeling the conditional distribution $p(y|x)$ with the goal of direct prediction of $y$ for each instance $x$. For prediction tasks, discriminative approaches are often more accurate since they use the model parameters more efficiently than generative approaches. Most of the current causal inference methods are discriminative, including the Balancing Neural Network (BNN) (Johansson et al., 2016), CounterFactual Regression Network (CFR-Net) (Shalit et al., 2017), and CFR-Net's extensions — *cf.*, (Yao et al., 2018; Hassanpour & Greiner, 2019; 2020) — as well as Dragon-Net (Shi et al., 2019).

**Generative methods**, on the other hand, describe the relationship between $x$ and $y$ by their joint probability distribution $p(x, y)$. This, in turn, would allow the generative model to answer arbitrary queries, including coping with missing features $x$ using the marginal distribution $p(x)$ or [similar to discriminative models] predicting the unknown target values $y$ via $p(y|x)$. A promising direction forward for causal inference is developing *generative* models, using either Generative Adverserial Network (GAN) (Goodfellow et al., 2014) or Variational Auto-Encoder (VAE) (Kingma & Welling, 2014; Rezende et al., 2014). This has led to two generative approaches for causal inference: GANs for inference of Individualised Treatment Effects (GANITE) (Yoon et al., 2018) and Causal Effect

VAE (CEVAE) Louizos et al. (2017). However, neither of the two achieve competitive performance in terms of treatment effect estimation compared to the discriminative approaches.

Although discriminative models have excellent predictive performance, they suffer from two drawbacks: (i) overfitting, and (ii) making highly-confident predictions, even for instances that are "far" from the observed training data. Generative models based on Bayesian inference, on the other hand, can handle both of these drawbacks: issue (i) can be minimized by taking an average over the posterior distribution of model parameters; and issue (ii) can be addressed by explicitly providing model uncertainty via the posterior (Gordon & Hernández-Lobato, 2020). Although the exact inference is often intractable, efficient approximations to the parameter posterior distribution is possible through variational methods. Here, we use the Variational Auto-Encoder (VAE) (Kingma & Welling, 2014; Rezende et al., 2014) for the Bayesian inference component of our causal inference method.

**Contribution:** In this paper, we propose three interrelated Bayesian model architectures (namely Series, Parallel, and Hybrid) that employ the VAE framework to address the task of causal inference for binary treatments. We find that the best performing architecture is the Hybrid model, that is [partially] successful in decomposing the underlying factors of any observational dataset. This is a valuable property, as that means it can accurately estimate all all treatment outcomes. We demonstrate that these models significantly outperform the state-of-the-art in terms of treatment effect estimation performance on two publicly available benchmarks, as well as a fully synthetic dataset that allows for detailed performance analyses.

## 2 RELATED WORKS

**CFR-Net**    Shalit et al. (2017) considered the binary treatment task and attempted to learn a representation space $\Phi$ that reduces selection bias by making $\Pr(\Phi(x) \,|\, t = 0)$ and $\Pr(\Phi(x) \,|\, t = 1)$ as close to each other as possible, provided that $\Phi(x)$ retains enough information that the learned regressors $\{h^t(\Phi(\cdot)) : t \in \{0, 1\}\}$ can generalize well on the observed outcomes. Their objective function includes $L[y_i, h^{t_i}(\Phi(x_i))]$, which is the loss of predicting the observed outcome for sample $i$ (described as $x_i$), weighted by $\omega_i = \frac{t_i}{2u} + \frac{1 - t_i}{2(1 - u)}$, where $u = \Pr(t = 1)$. This is effectively setting $\omega_i = \frac{1}{2\Pr(t_i)}$ where $\Pr(t_i)$ is the probability of selecting treatment $t_i$ over the entire population.

**DR-CFR**    Hassanpour & Greiner (2020) argued against the standard implicit assumption that *all* of the covariates $X$ are confounders (*i.e.*, contribute to both treatment assignment and outcome determination). Instead, they proposed a graphical model similar to that in Figure 1 and designed a discriminative causal inference approach accordingly — built on top of the CFR-Net. Specifically, their model, named Disentangled Representations for CFR (DR-CFR), includes three representation networks, each trained with constraints to insure that each component corresponds to its respective underlying factor. While the idea behind DR-CFR provides an interesting intuition, it is known that only generative models (and not discriminative ones) can truly identify the underlying data generating mechanism. This paper is a step in this direction.

**Dragon-Net**    Shi et al. (2019)'s main objective was to estimate the Average Treatment Effect (ATE), which they explain requires a two stage procedure: (i) fit models that predict the outcomes for both treatments; and (ii) find a downstream estimator of the effect. Their method is based on a classic result from strong ignorability — *i.e.*, Theorem 3 in (Rosenbaum & Rubin, 1983) — that states:

$$(y^1, y^0) \perp\!\!\!\perp t \,|\, x \qquad \& \qquad \Pr(t = 1 \,|\, x) \in (0, 1) \qquad \Longrightarrow$$
$$(y^1, y^0) \perp\!\!\!\perp t \,|\, b(x) \qquad \& \qquad \Pr(t = 1 \,|\, b(x)) \in (0, 1)$$

where $b(x)$ is a balancing score[1]. They consider propensity score as a balancing score and argue that only the parts of $X$ relevant for predicting $T$ are required for the estimation of the causal effect[2]. This theorem only provides a way to *match* treated and control instances though — *i.e.*, it helps finding potential counterfactuals from the alternative group to calculate ATE. Shi et al. (2019), however, used this theorem to derive minimal representations on which to *regress* to estimate the outcomes.

---

[1]That is, $X \perp\!\!\!\perp T \,|\, b(X)$    (Rosenbaum & Rubin, 1983).

[2]The authors acknowledge that this would hurt the predictive performance for individual outcomes. As a result, this yields inaccurate estimation of Individual Treatment Effects (ITEs).

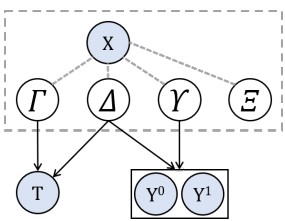
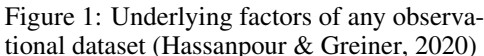

Figure 1: Underlying factors of any observational dataset (Hassanpour & Greiner, 2020)

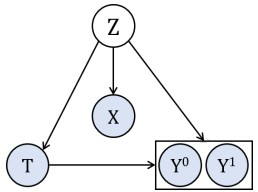

Figure 2: Graphical model of CEVAE (Louizos et al., 2017)

**GANITE**    Yoon et al. (2018) proposed the counterfactual GAN, whose generator $G$, given $\{x, t, y^t\}$, estimates the counterfactual outcomes ($\hat{y}^{\neg t}$); and whose discriminator $D$ tries to identify which of $\{[x, 0, y^0], [x, 1, y^1]\}$ is the factual outcome. It is, however, unclear why this requires that $G$ must produce samples that are indistinguishable from the factual outcomes, especially as $D$ can just learn the *treatment selection mechanism* instead of distinguishing the factual outcomes from counterfactuals. Although this work is among the few generative approaches for causal inference, our empirical results (in Section 4) show that it does not effectively estimate counterfactual outcomes.

**CEVAE**    Louizos et al. (2017) used VAE to extract latent confounders from their observed proxies in $X$. While this is an interesting step in the right direction, empirical results show that it does not always accurately estimate treatment effect (see Section 4). The authors note that this may be because CEVAE is not able to address the problem of selection bias. Another reason that we think contributes to CEVAE's sub-optimal performance is its assumed graphical model of the underlying data generating mechanism (depicted in Figure 2). This model assumes that there is only one latent variable $Z$ (confounding $T$ and $Y$) that generates the entire observational data; however, we know from (Kuang et al., 2017) and (Hassanpour & Greiner, 2020) that there must be more (see Figure 1).

[R2:] **M1 and M2 VAEs**    In an attempt to enhance the conventional representation learning with VAEs — referred to as the M1 model (Kingma & Welling, 2014; Rezende et al., 2014) — in a semi-supervised manner, Kingma et al. (2014) proposed the M2 VAE. While the M1 model helps learning latent representations from the covariate matrix $X$ alone, the M2 model allows the target information also to guide the representation learning process. In our work, the target information includes the treatment bit $T$ as well as the observed outcome $Y$. This additional information helps learning more expressive representations, that was not possible with the unsupervised M1 model. Appendix A.1 presents a more detailed overview of the M1 and M2 VAEs.

## 3    METHOD

Following (Hassanpour & Greiner, 2020) and without loss of generality, we assume that the random variable $X$ follows an unknown joint probability distribution $\Pr(X \mid \Gamma, \Delta, \Upsilon, \Xi)$, where $\Gamma$, $\Delta$, $\Upsilon$, and $\Xi$ are non-overlapping independent factors. Moreover, we assume that treatment $T$ follows $\Pr(T \mid \Gamma, \Delta)$ (*i.e.*, $\Gamma$ and $\Delta$ are the responsible factors for selection bias) and outcome $Y^T$ follows $\Pr_T(Y^T \mid \Delta, \Upsilon)$; see Figure 1. Observe that the factor $\Gamma$ (resp., $\Upsilon$) partially determines only $T$ (resp., $Y$), but not $Y$ (resp., $T$); and $\Delta$ includes the confounding factors between $T$ and $Y$.

Our goal is to design generative model architectures that encourage learning disentangled representations of these four underlying latent factors (see Figure 1). In other words, it is an attempt to decompose and separately learn the underlying factors that are responsible for determining $T$ and $Y$. To achieve this, we propose three architectures (as illustrated in Figures 3(a), 3(b), and 3(c)), each employing a VAE (Kingma & Welling, 2014; Rezende et al., 2014) that each include a decoder (generative model) and an encoder (variational posterior). Specifically, we use the M1 and M2 models from (Kingma et al., 2014) as our building blocks, leading to a Series architecture, a Parallel architecture, and a Hybrid one. Each component is parametrized as a deep neural network.

### 3.1    THE VARIATIONAL AUTO-ENCODER COMPONENT

#### 3.1.1    THE SERIES ARCHITECTURE

The architecture of the Series model is illustrated in Figure 3(a). Louizos et al. (2015) proposed a similar architecture to address fairness in machine learning, but using a binary sensitive variable $S$

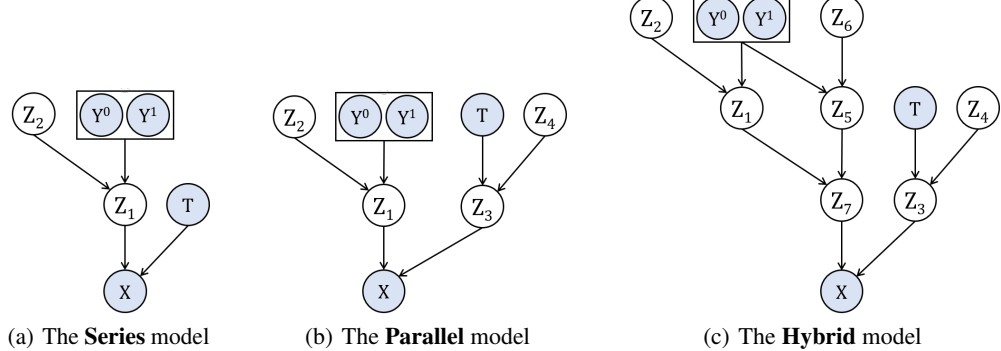

(a) The **Series** model      (b) The **Parallel** model      (c) The **Hybrid** model

Figure 3: Belief nets of the proposed architectures.

(*e.g.*, gender, race, etc.) rather than the treatment $T$. Here, we employ this architecture for causal inference and explain why it should work. [R4:] We hypothesize that this structure functions as a distillation tower: the bottom M2 VAE attempts to decompose $\Gamma$ (guided by $T$) from $\Delta$ and $\Upsilon$ (captured by $Z_1$); and the top M2 VAE attempts to learn $\Delta$ and $\Upsilon$ (guided by $Y$). Decoder and encoder components of the Series model (parametrized by $\theta_s$ and $\phi_s$ respectively) involve the following distributions:

| Priors | Likelihood | Posteriors |
|---|---|---|
| $p_{\theta_s}(z_2)$ | $p_{\theta_s}(x\|z_1, t)$ | $q_{\phi_s}(z_1\|x, t)$ |
| $p_{\theta_s}(z_1\|y, z_2)$ | | $q_{\phi_s}(y\|z_1)$ |
| | | $q_{\phi_s}(z_2\|y, z_1)$ |

The goal is to maximize the conditional log-likelihood of the observed data (left-hand-side of the following inequality) by maximizing the Evidence Lower BOund (ELBO; right-hand-side) — *i.e.*,

$$\sum_{i=1}^{N} \log p(x_i|t_i, y_i) \geq \sum_{i=1}^{N} \mathbb{E}_{q_{\phi_s}(z_1|x,t)}\big[\log p_{\theta_s}(x_i|z_{1i}, t_i)\big] \tag{1}$$

$$- \text{KL}\big(q_{\phi_s}(z_1|x, t) \,\|\, p_{\theta_s}(z_1|y, z_2)\big) - \text{KL}\big(q_{\phi_s}(z_2|y, z_1) \,\|\, p_{\theta_s}(z_2)\big) \tag{2}$$

where KL denotes the Kullback-Leibler divergence, $p_{\theta_s}(z_2)$ is the unit multivariate Gaussian (*i.e.*, $\mathcal{N}(0, \mathbb{I})$), and the other distributions are parameterized as deep neural networks.

### 3.1.2 THE PARALLEL ARCHITECTURE

[R4:] The Series model is composed of two M2 stacked models. However, Kingma et al. (2014) showed that an M1+M2 stacked architecture learns better representations than an M2 model alone for a downstream prediction task. This motivated us to design a double M1+M2 Parallel model; where one arm is for the outcome to guide the representation learning via $Z_1$ and another for the treatment to guide that via $Z_3$. This architecture is illustrated in Figure 3(b). We hypothesize that $Z_1$ would learn $\Delta$ and $\Upsilon$, and $Z_3$ would learn $\Gamma$ (and perhaps partially $\Delta$). Decoder and encoder components of the Parallel model (parametrized by $\theta_p$ and $\phi_p$ respectively) involve the following distributions:

| Priors | | Likelihood | Posteriors | |
|---|---|---|---|---|
| $p_{\theta_p}(z_2)$ | $p_{\theta_p}(z_4)$ | $p_{\theta_p}(x\|z_1, z_3)$ | $q_{\phi_p}(z_1\|x, t)$ | $q_{\phi_p}(z_3\|x, y)$ |
| $p_{\theta_p}(z_1\|y, z_2)$ | $p_{\theta_p}(z_3\|t, z_4)$ | | $q_{\phi_p}(y\|z_1)$ | $q_{\phi_p}(t\|z_3)$ |
| | | | $q_{\phi_p}(z_2\|y, z_1)$ | $q_{\phi_p}(z_4\|t, z_3)$ |

Here, the conditional log-likelihood can be upper bounded by:

$$\sum_{i=1}^{N} \log p(x_i|t_i, y_i) \geq \sum_{i=1}^{N} \mathbb{E}_{q_{\phi_p}(z_1, z_3|x,t,y)}\big[\log p_{\theta_p}(x_i|z_{1i}, z_{3i})\big] \tag{3}$$

$$- \text{KL}\big(q_{\phi_p}(z_1|x, t) \,\|\, p_{\theta_p}(z_1|y, z_2)\big) - \text{KL}\big(q_{\phi_p}(z_2|y, z_1) \,\|\, p_{\theta_p}(z_2)\big) \tag{4}$$

$$- \text{KL}\big(q_{\phi_p}(z_3|x, y) \,\|\, p_{\theta_p}(z_3|t, z_4)\big) - \text{KL}\big(q_{\phi_p}(z_4|t, z_3) \,\|\, p_{\theta_p}(z_4)\big) \tag{5}$$

### 3.1.3 THE HYBRID ARCHITECTURE

[R4:] The final architecture, Hybrid, attempts to get the best capabilities of the previous two architectures. The backbone of the Hybrid model has a Series architecture, that separates $\Gamma$ (factors related to the treatment $T$; captured by the right module with $Z_3$ as its head), from $\Delta$ and $\Upsilon$ (factors related to the outcome $Y$; captured by the left module with $Z_7$ as its head). The left module, itself, consists of a Parallel model that attempts to proceed one step further and decompose $\Delta$ from $\Upsilon$. This is done with the help of a discrepancy penalty (see Section 3.3). Figure 3(c) illustrates our designed architecture for the Hybrid model. Decoder and encoder components of the Hybrid model (parametrized by $\theta_h$ and $\phi_h$ respectively) involve the following distributions:

| Priors | | | Likelihood | Posteriors | | |
|---|---|---|---|---|---|---|
| $p_{\theta_h}(z_2)$ | $p_{\theta_h}(z_4)$ | $p_{\theta_h}(z_6)$ | $p_{\theta_h}(x|z_3, z_7)$ | $q_{\phi_h}(z_7|x,t)$ | $q_{\phi_h}(z_1|z_7)$ | $q_{\phi_h}(z_5|z_7)$ |
| $p_{\theta_h}(z_1|y,z_2)$ | $p_{\theta_h}(z_3|t,z_4)$ | $p_{\theta_h}(z_5|y,z_6)$ | | $q_{\phi_h}(z_3|x,y)$ | $q_{\phi_h}(y|z_1,z_5)$ | $q_{\phi_h}(t|z_3)$ |
| $p_{\theta_h}(z_7|z_1,z_5)$ | | | | $q_{\phi_h}(z_2|y,z_1)$ | $q_{\phi_h}(z_6|y,z_5)$ | $q_{\phi_h}(z_4|t,z_3)$ |

Here, the conditional log-likelihood can be upper bounded by:

$$\sum_{i=1}^{N} \log p(x_i|t_i, y_i) \geq \sum_{i=1}^{N} \mathbb{E}_{q_{\phi_h}(z_3, z_7|x,t,y)} \big[ \log p_{\theta_h}(x_i|z_{3i}, z_{7i}) \big] \tag{6}$$

$$- \text{KL}\big( q_{\phi_h}(z_1|z_7) \,||\, p_{\theta_h}(z_1|y,z_2) \big) - \text{KL}\big( q_{\phi_h}(z_2|y,z_1) \,||\, p_{\theta_h}(z_2) \big) \tag{7}$$

$$- \text{KL}\big( q_{\phi_h}(z_3|x,y) \,||\, p_{\theta_h}(z_3|t,z_4) \big) - \text{KL}\big( q_{\phi_h}(z_4|t,z_3) \,||\, p_{\theta_h}(z_4) \big) \tag{8}$$

$$- \text{KL}\big( q_{\phi_h}(z_5|z_7) \,||\, p_{\theta_h}(z_5|y,z_6) \big) - \text{KL}\big( q_{\phi_h}(z_6|y,z_5) \,||\, p_{\theta_h}(z_6) \big) \tag{9}$$

$$- \text{KL}\big( q_{\phi_h}(z_7|x,t) \,||\, p_{\theta_h}(z_7|z_1,z_5) \big) \tag{10}$$

The first term in the ELBO (*i.e.*, right-hand-side of Equations (1), (3), or (6)) is called the Reconstruction Loss (RecL) and the next term(s) (*i.e.*, Equation (2), summation of Equations (4) and (5), or summation of Equations (7), (8), (9), and (10)) is referred to as the KL [R1:] Divergence (KLD). Concisely, the ELBO can be written as: RecL $-$ KLD, which is to be maximized.

### 3.2 DISENTANGLEMENT WITH $\beta$-VAE

As mentioned earlier, we want the learned latent variables to be disentangled, to match our assumption of non-overlapping factors $\Gamma$, $\Delta$, and $\Upsilon$. To ensure this, we employ the $\beta$-VAE (Higgins et al., 2017) which adds a hyperparameter $\beta$ as a multiplier of the KLD part of the ELBO. This adjustable hyperparameter facilitates a trade-off that helps balance the latent channel capacity and independence constraints (handled by the KL terms) with the reconstruction accuracy — *i.e.*, including the $\beta$ hypararameter grants a better control over the level of disentanglement in the learned representations (Burgess et al., 2018). Therefore, the generative objective to be minimized becomes:

$$\mathcal{L}_{\text{VAE}} \quad = \quad -\text{RecL} \, + \, \beta \cdot \text{KLD} \tag{11}$$

Although Higgins et al. (2017) suggest the $\beta$ to be set greater than 1 in most applications, Hoffman et al. (2017) show that having a $\beta < 1$ weight on the KL term can be interpreted as optimizing the ELBO under an alternative prior, which functions as a regularization term to prevent degeneracy.

### 3.3 DISCREPANCY

Although all the three proposed graphical models suggest that $T$ and $Z_1$ are statistically independent (see, for example, the collider structure (at $X$): $T \to X \leftarrow Z_1$ in Figure 3(a)), an information leak is quite possible due to the correlation between the outcome $y$ and treatment $t$ in the data. We therefore require an extra regularization term on $q_\phi(z_1|t)$ in order to penalize the discrepancy (denoted by `disc`) between the conditional distributions of $z_1$ given $t = 0$ versus given $t = 1$. To achieve this regularization, we calculate the `disc` using an Integral Probability Metric (IPM) (Mansour et al., 2009) [c1] that measures the distance between the two above-mentioned distributions:

$$\mathcal{L}_{\text{disc}} \quad = \quad \text{IPM}\big( \{z_1\}_{i:t_i=0}, \, \{z_1\}_{i:t_i=1} \big) \tag{12}$$

[c1]In this work, we use the Maximum Mean Discrepancy (MMD) (Gretton et al., 2012).

### 3.4 PREDICTIVE LOSS

Note, however, that neither the VAE nor the `disc` losses contribute to training a predictive model for outcomes. To remedy this, we extend the objective function to include a discriminative term for the regression loss of predicting $y$: [c1]

$$\mathcal{L}_{\text{pred}} \quad = \quad \frac{1}{N} \sum_{i=1}^{N} \omega_i \cdot \mathcal{L}\big[\,y_i,\,\hat{y}_i\,\big] \tag{13}$$

where the predicted outcome $\hat{y}_i$ is derived as the mean of the $q_\phi^{t_i}(y_i|z_{1\,i})$ posterior trained for the respective treatment $t_i$; $\mathcal{L}\big[\,y_i,\,\hat{y}_i\,\big]$ is the factual loss (*i.e.*, L2 loss for real-valued outcomes and log loss for binary-valued outcomes); and $\omega_i$ represent the weights [R1:] that attempt to account for selection bias. We consider two approaches in the literature to derive the weights: (i) the *Population-Based* (PB) weights as proposed in CFR-Net (Shalit et al., 2017); and (ii) the *Context-Aware* (CA) weights as proposed by Hassanpour & Greiner (2019). Note that disentangling $\Delta$ from $\Upsilon$ is only beneficial when using the CA weights, since we need just the $\Delta$ factors to derive them (Hassanpour & Greiner, 2020).

### 3.5 FINAL MODEL(S)

Putting everything together, the overall objective function to be minimized is then:

$$\mathcal{J} \quad = \quad \mathcal{L}_{\text{pred}} \,+\, \alpha \cdot \mathcal{L}_{\texttt{disc}} \,+\, \gamma \cdot \mathcal{L}_{\text{VAE}} \,+\, \lambda \cdot \mathfrak{Reg} \tag{14}$$

where $\mathfrak{Reg}$ penalizes the model complexity.

This objective function is motivated by the work of McCallum et al. (2006), which suggested optimizing a convex combination of discriminative and generative losses would indeed improve predictive performance. As an empirical verification, note that for $\gamma = 0$, the Series and Parallel models effectively reduce to CFR-Net. However, our empirical results (*cf.*, Section 4) suggest that the generative term in the objective function helps learning representations that embed more relevant information for estimating outcomes than that of $\Phi$ in CFR-Net.

We refer to the family of our proposed methods as VAE-CI (Variational Auto-Encoder for Causal Inference); specifically: **{S, P, H}-VAE-CI**, for **S**eries, **P**arallel, and **H**ybrid respectively.

## 4 EXPERIMENTS, RESULTS, AND DISCUSSION

### 4.1 BENCHMARKS

**Infant Health and Development Program (IHDP)**     The original IHDP randomized controlled trial was designed to evaluate the effect of specialist home visits on future cognitive test scores of premature infants. Hill (2011) induced selection bias by removing a non-random subset of the treated population. The dataset contains 747 instances (608 control and 139 treated) with 25 covariates. We use the same benchmark (with 100 realizations of outcomes) provided by and used in (Johansson et al., 2016) and (Shalit et al., 2017).

**Atlantic Causal Inference Conference 2018 (ACIC'18)**     ACIC'18 is a collection of binary-treatment datasets released for a data challenge. Following (Shi et al., 2019), we use a subset of the datasets with instances $N \in \{1, 5, 10\} \times 10^3$ (four datasets in each category). The covariates matrix for each dataset involves 177 features and is sub-sampled from a table of medical measurements taken from the Linked Birth and Infant Death Data (LBIDD) (MacDorman & Atkinson, 1998), that contains information corresponding to 100,000 subjects.

**Fully Synthetic Datasets**     We generated a set of synthetic datasets according to the procedure described in (Hassanpour & Greiner, 2020); see Section A.2 for an overview. We considered all the viable datasets in a mesh generated by various sets of variables, of sizes $m_\Gamma, m_\Delta, m_\Upsilon \in \{0, 4, 8\}$ and $m_\Xi = 1$. This creates 24 scenarios[c1] that consider all possible relative sizes of the factors $\Gamma, \Delta,$

---

[c1]This is similar to the way (Kingma et al., 2014) included a classification loss in their Equation (9).

[c1]There are $3^3 = 27$ combinations in total; however, we removed three of these combinations that generate pure noise outcomes — *i.e.*, $\Delta = \Upsilon = \emptyset$: $(0, 0, 0)$, $(4, 0, 0)$, and $(8, 0, 0)$.

| H-VAE-CI | Z1 | Z2 | Z3 | Z4 | Z5 | Z6 | Z7 |
|---|---|---|---|---|---|---|---|
| Γ | 0.2262 | 0.2599 | **1.2512** | 1.1979 | 0.3109 | 0.2890 | 0.2959 |
| Δ | 0.6431 | 0.7704 | 0.5202 | 0.5602 | **0.8229** | 0.7073 | **0.7170** |
| Υ | **0.8957** | 0.7929 | 0.2622 | 0.2641 | 0.6551 | 0.5679 | **0.7106** |

| S-VAE-CI | Z1 | Z2 |
|---|---|---|
| Γ | 0.3407 | 0.3451 |
| Δ | **0.7925** | 0.7828 |
| Υ | **0.8804** | 0.8204 |

| P-VAE-CI | Z1 | Z2 | Z3 | Z4 |
|---|---|---|---|---|
| Γ | 0.2920 | 0.2922 | **0.6374** | 0.6676 |
| Δ | **0.7799** | 0.7563 | 0.1882 | 0.1888 |
| Υ | **0.7686** | 0.7735 | 0.4738 | 0.3737 |

| DR-CFR | Γ_rep | Δ_rep | Υ_rep |
|---|---|---|---|
| Γ | **0.5289** | 0.1733 | 0.8412 |
| Δ | **0.5429** | **0.7038** | 0.8402 |
| Υ | 0.0544 | **0.6749** | 1.0008 |

Figure 4: Performance analysis for decomposition of the underlying factors on the **synthetic** dataset with $m_{\Gamma,\Delta,\Upsilon} = 8, m_{\Xi} = 1$.

Table 1: **IHDP** (100 realizations) benchmark

| METHOD | PEHE | $\epsilon_{ATE}$ |
|---|---|---|
| CFR | 0.75 (0.57) | 0.08 (0.10) |
| DR-CFR | 0.65 (0.37) | 0.03 (0.04) |
| DRAGON | NA | 0.14 (0.15) |
| GANITE | 2.81 (2.30) | 0.24 (0.46) |
| CEVAE | 2.50 (3.47) | 0.18 (0.25) |
| S-VAE-CI | **0.51** (0.37) | **0.00** (0.02) |
| P-VAE-CI | **0.52** (0.36) | **0.01** (0.03) |
| H-VAE-CI (PB) | **0.49** (0.36) | **0.01** (0.02) |
| H-VAE-CI (CA) | **0.48** (0.35) | **0.01** (0.01) |

Table 2: **ACIC'18** ($N \leq 10K$) benchmark

| METHOD | PEHE | $\epsilon_{ATE}$ |
|---|---|---|
| CFR | 5.13 (5.59) | 1.21 (1.81) |
| DR-CFR | 3.86 (3.39) | 0.80 (1.41) |
| DRAGON | NA | 0.48 (0.77) |
| GANITE | 3.55 (2.27) | 0.69 (0.65) |
| CEVAE | 5.30 (5.52) | 3.29 (3.50) |
| S-VAE-CI | 2.73 (2.39) | 0.51 (0.82) |
| P-VAE-CI | 2.62 (2.26) | 0.37 (0.75) |
| H-VAE-CI (PB) | 1.78 (1.27) | 0.44 (0.77) |
| H-VAE-CI (CA) | 1.66 (1.30) | 0.39 (0.75) |

PEHE and $\epsilon_{ATE}$ measures (lower is better) represented in the form of "mean (standard deviation)".

and $\Upsilon$. For each scenario, we synthesized multiple datasets with various initial random seeds in order to allow for statistical significance testing of the performance comparisons between various methods.

## 4.2 Evaluating Identification of the Underlying Factors

To evaluate the identification performance of the underlying factors, we use a fully synthetic dataset with $m_{\Gamma} = m_{\Delta} = m_{\Upsilon} = 8$ and $m_{\Xi} = 1$. We set $x$ to be one of the four dummy vectors $V_{1..4}$ and input it to each trained representation network $Z_j$. Three of these vectors had "1" in the 8 positions associated with $\Gamma$, $\Delta$, and $\Upsilon$ respectively, and the remaining 17 positions of each vector was filled with "0". The fourth vector was all "1" except for the last position (the noise) which was "0". This helps measure the maximum amount of information that is passed to the final layer of each representation network.

We let $O_{i,j}$ be the `elu` output (here, $\in \mathbb{R}^{200}$) of the encoder network $Z_j$ when $x = V_i$. The average of the 200 values of $O_{i,j}$ ($Avg(O_{i,j})$) represents the power of signal that was produced by the $Z_j$ channel on the input $V_i$. The values shown in Figure 4's tables are the ratios of $Avg(O_{1,j})$, $Avg(O_{2,j})$, and $Avg(O_{3,j})$ divided by $Avg(O_{4,j})$ for each of the learned representation networks. Note that, a larger ratio indicates that the respective representation network $Z_j$ has allowed more of the input signal $V_i$ to pass through. Section A.3 includes more details on this procedure.[c0]

As expected, $Z_3$ and $Z_4$ capture $\Gamma$ (*e.g.*, the $Z_3$ ratios for $\Gamma$ in the {P, H}-VAE-CI tables are largest), and $Z_1$, $Z_2$, $Z_5$, $Z_6$, and $Z_7$ capture $\Delta$ and $\Upsilon$. Note that decomposition of $\Delta$ from $\Upsilon$ has not been achieved by any of the methods except for H-VAE-CI, which captures $\Upsilon$ by $Z_1$ and $\Delta$ by $Z_5$ [R4:] (note the ratios are largest for $Z_1$ and $Z_5$). This decomposition is vital for deriving context-aware importance sampling weights because they must be calculated from $\Delta$ only (Hassanpour & Greiner, 2020). Also observe that both {P, H}-VAE-CI are able to separate $\Gamma$ from $\Delta$. However, DR-CFR, which tried to disentangle all factors, failed not only to disentangle $\Delta$ from $\Upsilon$, but also $\Gamma$ from $\Delta$.

## 4.3 Evaluating Treatment Effect Estimation

Evaluation of treatment effect estimation is often done with semi- or fully- synthetic datasets that include both factual and counterfactual outcomes. There are two categories of performance measures:

---

[c0]Unlike the evaluation strategy presented in (Hassanpour & Greiner, 2020) that only looked at the first layer's weights of each representation network, we propagate the values through the entire network and check how much of each factor is exhibited in the final layer of every representation network. [R1 & R4:] Yet, the proposed procedure still crudely evaluates the quality of disentanglement of the underlying factors. We did explore using the Mutual Information (Belghazi et al., 2018) for this task (not shown here); however, it appears that it does not work for high-dimensional data such as ours. All in all, more research is needed to address this task.

**Individual-based:** "Precision in Estimation of Heterogeneous Effect" $\text{PEHE} = \sqrt{\frac{1}{N} \sum_{i=1}^{N} (\hat{e}_i - e_i)^2}$ uses $\hat{e}_i = \hat{y}_i^1 - \hat{y}_i^0$ as the estimated effect and $e_i = y_i^1 - y_i^0$ as the true effect (Hill, 2011); and

**Population-based:** "Bias of the Average Treatment Effect" $\epsilon_{\text{ATE}} = \left| \text{ATE} - \widehat{\text{ATE}} \right|$, where $\text{ATE} = \frac{1}{N} \sum_{i=1}^{N} y_i^1 - \frac{1}{N} \sum_{j=1}^{N} y_j^0$ and $\widehat{\text{ATE}}$ is calculated based on the estimated outcomes.

In this paper, we compare performances of our proposed methods **{S, P, H}-VAE-CI** versus the following treatment effect estimation methods: **CFR-Net** (Shalit et al., 2017), **DR-CFR** (Hassanpour & Greiner, 2020), **Dragon-Net** (Shi et al., 2019), **GANITE** (Yoon et al., 2018), and **CEVAE** (Louizos et al., 2017). The basic search grid for hyperparameters of the CFR-Net based algorithms (including our proposed methods) is available in Section A.4. For the other algorithms, we searched around their default hyperparameters' setting. We ran the experiments for the contender methods using their publicly available code-bases; note the following points:

- Since Dragon-Net is designed to estimate ATE only, we did not report its performance results for the PEHE measure (which, as expected, were significantly inaccurate).
- The original GANITE code-base was implemented for binary outcomes only. We modified the code (losses, etc.) such that it could process real-valued outcomes also.
- We were surprised that CEVAE diverged when running on the ACIC'18 datasets. To avoid this, we had to run the ACIC'18 experiments on the binary covariates only.

Tables 1, 2, and 3 summarize the mean and standard deviation of the PEHE and $\epsilon_{\text{ATE}}$ measures (lower is better) on the IHDP, ACIC'18, and Synthetic benchmarks respectively. VAE-CI achieves the best performance among the contending methods. These results are statistically significant (in **bold**; based on Welch's unpaired t-test with $\alpha = 0.05$) for the IHDP and Synthetic benchmarks. Although VAE-CI also achieves the best performance on the ACIC'18 benchmark, the results are not statistically significant due to the high standard deviation of the contending methods' performances.

Figure 5 visualizes the PEHE measures on the entire synthetic datasets with sample size of $N = 10{,}000$. We observe that both plots corresponding to H-VAE-CI method (PB as well as CA) are inscribed by plots of all other methods, showcasing H-VAE-CI's superior performance under every possible selection bias scenario. *R1:* Note that for scenarios where $m_\Delta = 0$ (*i.e.,* the ones of form $m_\Gamma\_0\_m_\Upsilon$ on perimeter of the radar chart in Figure 5), the performances of H-VAE-CI (PB) and H-VAE-CI (CA) are almost identical. This is expected, since for these scenarios, the learned representation for $\Delta$ would be degenerate, and therefore, the context-aware weights would reduce to population-based ones. On the other hand, for scenarios where $m_\Delta \neq 0$, the H-VAE-CI (CA) often performs better than H-VAE-CI (PB). This can be attributed to the fact that H-VAE-CI has correctly disentangled $\Delta$ from $\Upsilon$. This facilitates learning good context-aware weights that better account for selection bias, which in turn, results in a better causal effect estimation performance.

We also performed hyperparameters' sensitivity analyses in terms of PEHE (see Figure 6). We discuss the results in the following:

- For the $\alpha$ hyperparameter (*i.e.,* coefficient of the discrepancy penalty), Figure 6(a) suggests that DR-CFR and H-VAE-CI methods have the most robust performance *R1 & R4:* throughout various values of $\alpha$. This is expected, because, unlike CFR and {S, P}-VAE-CI, DR-CFR and H-VAE-CI possess an independent node for representing $\Delta$. This helps them still capture $\Delta$ as $\alpha$ grows; since for them, $\alpha$ only affects learning a representation of $\Upsilon$.

  Comparing H-VAE-CI (PB) with (CA), we observe that for all $\alpha > 0.01$, H-VAE-CI (CA) outperforms H-VAE-CI (PB). This is because the discrepancy penalty would force $Z_1$ to only capture $\Upsilon$ and $Z_5$ to only capture $\Delta$. This results in deriving better CA weights (that should be learned from $\Delta$; here, from its learned representation $Z_5$). H-VAE-CI (PB), on the other hand, cannot take advantage of this disentanglement, which explains its sub-optimal performance.

- According to Figure 6(b), various $\beta$ values (*i.e.,* coefficient of KL divergence penalty) *R1 & R3 & R4:* do not make much difference for H-VAE-CI (except for $\beta \geq 1$; since this large value means the learned representations will be close to Gaussian noise). We initially thought using $\beta$-VAE might help *further* disentangle the underlying factors. However, Figure 6(b) suggests that close-to-zero or even zero $\beta$s also work effectively. We now think that the H-VAE-CI's architecture itself sufficiently decomposes the $\Gamma$, $\Delta$, and $\Upsilon$ factors, without needing the help of a KLD penalty. Appendix A.5 includes more evidence and a detailed discussion on why this interpretation should hold.

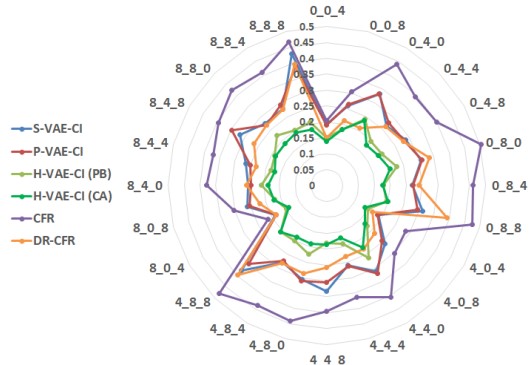

Figure 5: Radar graphs of PEHE (on the radii; lower is better) for the entire **synthetic** benchmark ($24 \times 3$ with $N = 10,000$; each vertex denotes the respective dataset). Figure is best viewed in color.

Table 3: PEHE and $\epsilon_{\text{ATE}}$ measures (lower is better) represented in the form of "mean (standard deviation)" on the entire **synthetic** benchmark (average performance of $24 \times 3$ datasets, each with sample size of 10,000).

| METHOD | PEHE | $\epsilon_{\text{ATE}}$ |
|---|---|---|
| **CFR** | 0.39 (0.08) | 0.027 (0.020) |
| **DR-CFR** | 0.26 (0.07) | 0.007 (0.004) |
| **DRAGON** | NA | 0.007 (0.005) |
| **GANITE** | 1.28 (0.43) | 0.036 (0.015) |
| **CEVAE** | 1.39 (0.32) | 0.287 (0.217) |
| **S-VAE-CI** | 0.28 (0.05) | 0.004 (0.003) |
| **P-VAE-CI** | 0.28 (0.05) | 0.004 (0.003) |
| **H-VAE-CI (PB)** | **0.20 (0.03)** | **0.003 (0.002)** |
| **H-VAE-CI (CA)** | **0.18 (0.02)** | **0.003 (0.002)** |

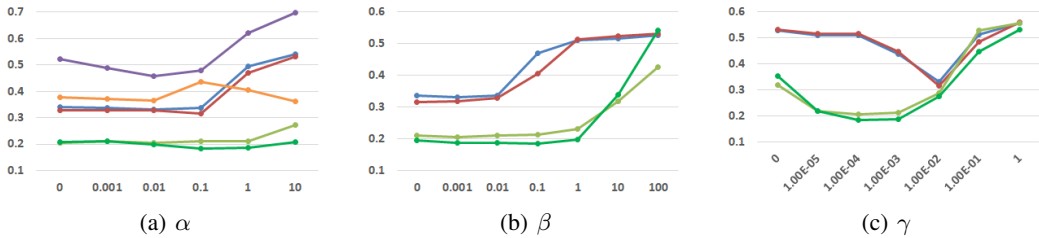

(a) $\alpha$        (b) $\beta$        (c) $\gamma$

Figure 6: Hyperparameters' ($x$-axis) sensitivity analysis based on PEHE ($y$-axis) on the **synthetic** dataset with $m_{\Gamma,\Delta,\Upsilon} = 8, m_{\Xi} = 1$. Legend is the same as Figure 5. Plots are best viewed in color.

- *R1 & R4:* For hyperparameter $\gamma$ (*i.e.*, coefficient of the generative loss penalty), H-VAE-CI achieves the most stable performance compared to the {S, P}-VAE-CI models — see Figure 6(c). Of particular interest is the superior performance of H-VAE-CI for $\gamma \leq 0.01$ compared to that of {S, P}-VAE-CI. This means that having the generative loss term (*i.e.*, $\mathcal{L}_{\text{VAE}}$) is more important for {S, P}-VAE-CI than for H-VAE-CI to perform well — note an extreme case happens at $\gamma = 0$, where the latter performs significantly better than the former. We hypothesize that this is because H-VAE-CI already learns expressive representations $Z_3$ and $Z_7$, meaning the optimization no longer really *requires* the $\mathcal{L}_{\text{VAE}}$ term to impose that. This is in contrast to $Z_1$ in S-VAE-CI and $Z_1$ and $Z_3$ in P-VAE-CI.

## 5    FUTURE WORKS AND CONCLUSION

Despite the success of the proposed methods, especially the Hybrid model, in addressing causal inference, no known algorithms can yet learn to perfectly disentangle factors $\Delta$ and $\Upsilon$. *R1:* This goal is important because we know isolating $\Delta$, and learning context-aware weights from it, does enhance the quality of the causal effect estimation performance — note the superior performance of H-VAE-CI (CA). *R1 & R3:* The results of our ablation study (in Figure 6(b)), however, revealed that the currently used $\beta$-VAE does not help with disentanglement of the underlying factors. This shows that we can attribute all the decomposition we get to the proposed architectures and objective function. A future direction of this research is to explore the use of better disentangling constraints — *e.g.*, works of Chen et al. (2018) and Lopez et al. (2018) — to see if they would yield sharper results.

The goal of this paper was to estimate treatment effects (either for individuals or the entire population) from observational data. We designed three VAE-based (Kingma & Welling, 2014; Rezende et al., 2014) architectures (namely Series, Parallel, and Hybrid), that employed (Kingma et al., 2014)'s M1 and M2 models. The Hybrid model, as the best performing architecture, partially succeeded at decomposing the underlying factors $\Gamma$, $\Delta$, and $\Upsilon$; which helped in accurate estimation of treatment outcomes. Our empirical results demonstrated the superiority of the proposed methods, compared to both state-of-the-art discriminative as well as generative approaches in the literature.

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

# A APPENDIX

## A.1 M1 AND M2 VARIATIONAL AUTO-ENCODERS

As the first proposed model, the M1 VAE is the conventional model that is used to learn representations of data (Kingma & Welling, 2014; Rezende et al., 2014). These features are learned from the covariate matrix $X$ only. Figure 7(a) illustrates the encoder and decoder of the M1 VAE. Note the graphical model on the left depicts the encoder; and the one on the right depict the decoder, which has arrows going the other direction.

Proposed by Kingma et al. (2014), the M2 model was an attempt to incorporate the information in target $Y$ into the representation learning procedure. This results in learning representations that separate specifications of individual targets from general properties shared between various targets. In case of digit generation, this translates into separating specifications that distinguish each digit from writing style or lighting condition. Figure 7(b) illustrates the encoder and decoder of the M2 VAE.

We can stack the M1 and M2 models as shown in Figure 7(c) to get the best results. This way, we can first learn a representation $Z_1$ from raw covariates, then find a second representation $Z_2$, now learning from $Z_1$ instead of the raw data.

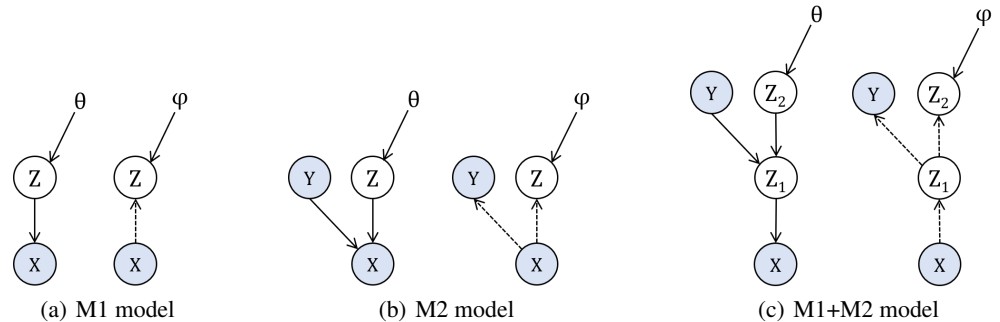

(a) M1 model          (b) M2 model          (c) M1+M2 model

Figure 7: Decoders (parametrized by $\theta$) and encoders (parametrized by $\phi$) of the M1, M2, and M1+M2 VAEs.

## A.2 PROCEDURE OF GENERATING THE SYNTHETIC DATASETS

Given as input the sample size $N$; dimensionalities $[m_\Gamma, m_\Delta, m_\Upsilon] \in \mathcal{Z}^{+(3)}$; for each factor $L \in \{\Gamma, \Delta, \Upsilon\}$, the means and covariance matrices $(\mu_L, \Sigma_L)$; and a scalar $\zeta$ that determines the slope of the logistic curve.

- For each latent factor $L \in \{\Gamma, \Delta, \Upsilon\}$, form $L$ by drawing $N$ instances (each of size $m_L$) from $\mathcal{N}(\mu_L, \Sigma_L)$. The covariates matrix $X$ is the result of concatenating $\Gamma$, $\Delta$, and $\Upsilon$. Refer to the concatenation of $\Gamma$ and $\Delta$ as $\Psi$ and that of $\Delta$ and $\Upsilon$ as $\Phi$ (for later use).
- For treatment $T$, sample $m_\Gamma + m_\Delta$ tuple of coefficients $\theta$ from $\mathcal{N}(0, 1)^{m_\Gamma + m_\Delta}$. Define the logging policy as $\pi_0(t=1 \,|\, z) = \frac{1}{1+\exp(-\zeta z)}$, where $z = \Psi \cdot \theta$. For each instance $x_i$, sample treatment $t_i$ from the Bernoulli distribution with parameter $\pi_0(t=1 \,|\, z_i)$.
- For outcomes $Y^0$ and $Y^1$, sample $m_\Delta + m_\Upsilon$ tuple of coefficients $\vartheta^0$ and $\vartheta^1$ from $\mathcal{N}(0, 1)^{m_\Delta + m_\Upsilon}$. Define $y^0 = (\Phi \circ \Phi \circ \Phi + 0.5) \cdot \vartheta^0 / (m_\Delta + m_\Upsilon) + \varepsilon$ and $y^1 = (\Phi \circ \Phi) \cdot \vartheta^1 / (m_\Delta + m_\Upsilon) + \varepsilon$, where $\varepsilon$ is a white noise sampled from $\mathcal{N}(0, 0.1)$ and $\circ$ is the symbol for element-wise product.

## A.3 EVALUATING IDENTIFICATION OF THE UNDERLYING FACTORS

Here, we elaborate on the procedure we followed to evaluate identification performance of the underlying factors. We produced four dummy vectors $V_i \in \mathbb{R}^{m_\Gamma + m_\Delta + m_\Upsilon + m_\Xi}$ as depicted on the left-side of Figure 8. The first to third vectors had ones (constant) in the positions associated with $\Gamma$,

$\Delta$, and $\Upsilon$ respectively, and the remainder of them were filled with zeroes. The fourth vector was all ones, so we can measure the maximum amount of information that is passed to the final layer of each representation network.

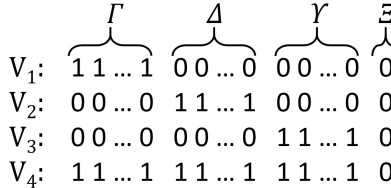 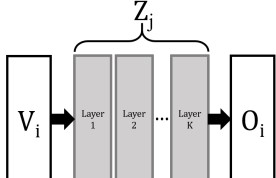

Figure 8: The four dummy $x$-like vectors (left); and the input/output vectors of the representation networks (right).

In the next step, each vector $V_i$ is fed to each trained network $Z_j$, and the output $O_i$ is recorded (see the right-side of Figure 8). The average of $O_i$ represents the power of signal that was communicated from $V_i$ and passed through the $Z_j$ channel. The values reported in the tables illustrated in Figure 4 are the ratios of {average of $O_1, O_2, O_3$} divided by {average of $O_4$} for all the learned representation networks.

## A.4 HYPERPARAMETERS

For all CFR, DR-CFR, and VAE-CI methods, we trained the neural networks with 3 layers (each consisting 200 hidden neurons)[c0], non-linear activation function `elu`, regularization coefficient of $\lambda$=1E-4, `Adam` optimizer (Kingma & Ba, 2015) with a learning rate of 1E-3, batch size of 300, and maximum number of iterations of $10,000$. See Table 4 for our hyperparameter search space.

Table 4: Hyperparameters and ranges

| Hyperparameter | Range |
|---|---|
| Discrepancy coefficient $\alpha$ | $\{0, 1E\{-3, -2, -1, 0, 1\}\}$ |
| KLD coefficient $\beta$ | $\{0, 1E\{-3, -2, -1, 0, 1, 2\}\}$ |
| Generative coefficient $\gamma$ | $\{0, 1E\{-5, -4, -3, -2, -1, 0\}\}$ |

## A.5 A DETAILED ANALYSIS OF THE EFFECT OF $\beta$

[R1 & R3]: Our initial hypothesis in using $\beta$-VAE was that it might help *further* disentangle the underlying factors, in addition to the other constraint already in place (*i.e.*, the architecture as well as the discrepancy penalty). However, Figure 6(b) suggests that close-to-zero or even zero $\beta$s also work effectively. To further explore this hypothesis, we examined the decomposition tables (similar to Figure 4) of H-VAE-CI for extreme configurations with $\beta = 0$ and observed that they were all effective at decomposing the underlying factors $\Gamma$, $\Delta$, and $\Upsilon$ (similar to the performance reported in the green table in Figure 4). Figure 9 shows several of these tables.

[R1 & R3]: Our interpretation of this observation is that the H-VAE-CI's architecture already takes care of decomposing the $\Gamma$, $\Delta$, and $\Upsilon$ factors, without needing the help of a KLD penalty. This means either of the following is happening: (i) $\beta$-VAE is not the best performing disentangling method and other disentangling constraints should be used instead — *e.g.*, works of Chen et al. (2018) and Lopez et al. (2018); or (ii) it is theoretically impossible to achieve disentanglement without some

---

[c0][R1 & R2]: In addition to this basic configuration, we also perform our grid search with an updated number of layers and/or number of neurons in each layer. This makes sure that all methods enjoy a similar model complexity.

| H-VAE-CI | Z1 | Z2 | Z3 | Z4 | Z5 | Z6 | Z7 |
|---|---|---|---|---|---|---|---|
| Γ | 0.2181 | 0.2185 | **1.8791** | **1.7711** | 0.2164 | 0.2190 | 0.2039 |
| Δ | 0.6041 | 0.6051 | 0.6142 | 0.6308 | **0.8688** | 0.8315 | **0.6138** |
| Y | **0.8523** | 0.8552 | 0.3321 | 0.3834 | 0.7552 | 0.7859 | 0.7384 |

| H-VAE-CI | Z1 | Z2 | Z3 | Z4 | Z5 | Z6 | Z7 |
|---|---|---|---|---|---|---|---|
| Γ | 0.2242 | 0.2182 | **0.7439** | 0.6770 | 0.2373 | 0.2583 | 0.2189 |
| Δ | 0.3014 | 0.2963 | 0.2612 | 0.3169 | **0.6051** | 0.5845 | **0.7048** |
| Y | **0.5385** | 0.5430 | 0.3211 | 0.3303 | 0.4394 | 0.4412 | **0.4571** |

| H-VAE-CI | Z1 | Z2 | Z3 | Z4 | Z5 | Z6 | Z7 |
|---|---|---|---|---|---|---|---|
| Γ | 0.4254 | 0.4493 | **0.7090** | 0.6872 | 0.3823 | 0.3771 | 0.3874 |
| Δ | 0.6438 | 0.6461 | 0.2750 | 0.3129 | **0.7452** | 0.7569 | **0.8237** |
| Y | **0.7760** | 1.1200 | 0.3137 | 0.3480 | 0.7240 | 0.7464 | **0.6717** |

| H-VAE-CI | Z1 | Z2 | Z3 | Z4 | Z5 | Z6 | Z7 |
|---|---|---|---|---|---|---|---|
| Γ | 0.3646 | 0.3643 | **1.1457** | 0.8659 | 0.3942 | 0.4069 | 0.3166 |
| Δ | 0.5127 | 0.5307 | 0.6463 | 0.5794 | **0.7016** | 0.6717 | **0.7652** |
| Y | **0.5565** | 0.5780 | 0.4260 | 0.3964 | 0.4119 | 0.4234 | **0.4534** |

| H-VAE-CI | Z1 | Z2 | Z3 | Z4 | Z5 | Z6 | Z7 |
|---|---|---|---|---|---|---|---|
| Γ | 0.8821 | 0.8752 | **0.5805** | 0.5782 | 0.3326 | 0.3309 | 0.4006 |
| Δ | 1.2542 | 1.2480 | 0.2843 | 0.3488 | **0.8553** | 0.8568 | **0.9392** |
| Y | **1.914** | 1.923 | 0.4498 | 0.4797 | 0.7791 | 0.7757 | **0.7969** |

| H-VAE-CI | Z1 | Z2 | Z3 | Z4 | Z5 | Z6 | Z7 |
|---|---|---|---|---|---|---|---|
| Γ | 0.0850 | 0.0875 | **1.8791** | **1.7711** | 0.1464 | 0.1459 | 0.1833 |
| Δ | 0.6107 | 0.6085 | 0.6142 | 0.6308 | **0.7851** | 0.7937 | **0.6878** |
| Y | **0.8349** | 0.8242 | 0.3321 | 0.3834 | 0.5177 | 0.5073 | **0.6832** |

Figure 9: Decomposition tables for H-VAE-CI with $\beta = 0$.

supervision (Locatello et al., 2019), which might not be possible to provide in this task. Exploring these options is out of the scope of this paper and is left to future work.

