# OpenReview forum: "Variational Auto-Encoder Architectures that Excel at Causal Inference"
_ICLR.cc/2021/Conference — Reject_

### Official Review · AnonReviewer1 · 2020-10-23
**The ablation studies are a bit concerning**

**Rating:** 6
**Confidence:** 4

**Review:**

The authors provide some novel VAE architectures for causal inference. They compare different architectures and assess their respective strengths and weaknesses. They empirically test the models on synthetic and real-world data and compare them to some baselines.

Major comments:
- The Hybrid model seems to perform best, but it also has the largest number of parameters and latent variables. What would happen if one would introduce even more layers of latent variables? Would the performance get even better? Is there a tradeoff between performance and complexity?
- In Figure 6 (a,b), it seems like the best method (H-VAE-CI) works very well with alpha=0 and beta=0. How can that be explained? If those loss terms are not needed, what is it that actually makes the model better?
- Moreover, in Fig. 6 (c), the H-VAE-CI also still has a considerable gap to the other models at gamma=0, which means completely without VAE loss. How can that be explained?

Minor comments:
- The KLD is the Kullback-Leibler *divergence*, which is crucially *not a distance* (because it's not symmetric)
- Could a more performant disentangling VAE be used? The beta-VAE is generally not the best [1]
- Generally, how do these disentanglement results relate to the impossibility theorem in [1]?
- How is the IPM chosen for L_disc? If it's the MMD, how is the kernel chosen? Does it depend on the task?
- In Table 3, the DR-CFR seems better than the S-VAE-CI and P-VAE-CI. Why are they boldened?

Summary:
The idea is very interesting, but the experiments are not fully convincing. Especially the ablation studies (Fig. 6) seem to suggest that most of the loss terms are not actually needed for the proposed model (H-VAE-CI) to perform well. I think further experimentation is needed to fully understand what really makes the proposed model work better than the baselines. For this, it could also be interesting to construct even "deeper" models than the (H-VAE-CI), which would be less informed by prior assumptions, and see if they might not work even better.

Update: Thanks to the additional experiment and discussions, I have increased my score.


[1] Locatello, F., Bauer, S., Lucic, M., Raetsch, G., Gelly, S., Schölkopf, B., & Bachem, O. (2019, May). Challenging common assumptions in the unsupervised learning of disentangled representations. In international conference on machine learning (pp. 4114-4124).

---

> ### Author Response · Authors · 2020-11-18
> **We have updated the paper to address your great questions / suggestions.**
>
> We thank R1 for the excellent, thought-provoking feedback and suggestions. Addressing those insightful questions resulted in a modified paper that includes strong analytical discussions and new exciting findings that substantially improved our paper. For this, we are very grateful!
>
>
> **Is there a tradeoff between performance and complexity?**
>
> We ran a version of CFR with deeper as well as wider networks, such that it has the exact same number of neurons as H-VAE-CI. For the Synthetic benchmark, the $\textrm{PEHE}$ improved only slightly, from $0.41 {\scriptsize(0.10)}$ to $0.39 {\scriptsize (0.08)}$, which is not statistically significant (based on Welch's unpaired t-test with $\alpha=0.05$); but the $\epsilon_{\textrm{ATE}}$ degraded from $0.027 {\scriptsize (0.020)}$ to $0.040 {\scriptsize (0.028)}$. For the IHDP and ACIC’18 benchmarks, the performance significantly degraded with bigger networks. This suggests a tradeoff between performance and complexity, and that the architecture and objective function of H-VAE-CI is indeed adding value compared to those of CFR. We have updated the performance results in Table 3 as well as Figure 5, and added a footnote in Appendix A.4 -- see the added text in orange and marked as “R1 & R2”.
>
>
> **Construct "deeper" models than the (H-VAE-CI) and see if they work even better.**
>
> Great suggestion! We ran the H-VAE-CI code with both deeper and wider networks on the Synthetic benchmark. However, going deeper with {layer_in=5, width_in=200} significantly worsened the performance. Going wider did not improve it either, as the $\textrm{PEHE}$ performance for the original network with {layer_in=3, width_in=200} is $0.20 {\scriptsize (0.03)}$ (as reported in the paper), which only matched the performance of wider networks with {layer_in=3, width_in=[350, 500]}: $0.20 {\scriptsize (0.05)}$.
>
>
> **Analyse results of the ablation study for $\alpha=0$.**
>
> For $\alpha=0$, $Z_1$ will no longer capture mostly $\Upsilon$ and $Z_5$ will not capture mostly $\Delta$; in other words, $Z_1$ and $Z_5$ will not learn disentangled factors. However, this has not negatively impacted the performance since the current implementation does not *require* this disentanglement to work properly. This is because the weighting scheme in our original submission (as stated at the end of Section 3.4) is *Population-Based* (PB) -- i.e., it is only based on $Pr(\,t\,)$; equivalent to the weighting scheme used in (Shalit et al., 2017). We now refer to this specific model as “H-VAE-CI (PB)”. Note that when the outcome prediction is based on a PB weighted factual loss, we only need the concatenation of $\Delta$ and $\Upsilon$. Therefore, even if the learned representations $Z_1$ and $Z_5$ are not disentangled, it will not impact the outcome prediction performance.
>
> Disentangling $\Delta$ from $\Upsilon$ only becomes important when we want to *further* account for selection bias using *Context-Aware* (CA) re-weighting of the factual loss (e.g., Hassanpour and Greiner (2019)’s work). As stated in Section 5, we had left the use of $\Delta$-based CA weights to future work. However, in the rebuttal period, we ran H-VAE-CI with Hassanpour and Greiner (2019)’s CA weighting scheme (as a proof of concept), leading to “H-VAE-CI (CA)”. As expected, we found out that the performance improves from $\textrm{PEHE}$ of $0.20 {\scriptsize (0.03)}$ to $0.18 {\scriptsize (0.02)}$ (this improvement is statistically significant) and no change in $\epsilon_{\textrm{ATE}}$ of $0.003 {\scriptsize (0.002)}$.
>
> On another note, observe that the performance of CFR, as well as {S,P}-VAE-CI, deteriorates as $\alpha$ increases from $0.1$. This is because this penalty term becomes so strong that $\Delta$ is removed from their $Z_1$, and since these methods do not have another node to capture $\Delta$, their outcome prediction performance is hindered. Note that neither DR-CFR nor H-VAE-CI suffer from this issue since they have another independent node to capture $\Delta$.
>
> To summarize, one important contribution of our work was showing that it is *possible* to learn disentangled representations of the underlying factors in observational studies. We only hypothesised that such disentanglement should be advantageous for causal effect estimation performance. However, our new results (that we produced during the rebuttal period) show that it is indeed *advantageous* to have disentangled representations if context-aware importance weighting is used. **Thanks R1 for your excellent question!** We have updated the paper to include these new exciting results. See the updated text in purple in pages 6 and 8 (marked as “R1”) and green text in page 8 (marked as “R1 & R4”), as well as updated Tables 1, 2, and 3, and Figures 5 and 6.

---

> > ### Author Response · Authors · 2020-11-18
> > **Rebuttal continued ...**
> >
> > **Analyse results of the ablation study for $\beta=0$; whether a more performant disentangling VAE could be used; and relationship to the impossibility theorem.**
> >
> > Regarding $\beta$ values, our hypothesis was that using $\beta$-VAE might help *further* disentangle the underlying factors; however, as you correctly point out, it appears that close-to-zero or even zero $\beta$s work perfectly fine. Our empirical results support this observation also, as we looked at the decomposition tables (similar to those in Figure 4) of H-VAE-CI for configurations with $\beta=0$ and observed that they all did a fairly good job (similar to the performance reported in the green table in Figure 4) at decomposing the underlying factors $\Gamma$, $\Delta$, and $\Upsilon$. The newly added Figure 9 in the Appendix shows several of these tables.
> >
> > Our interpretation of this observation is that the H-VAE-CI architecture is already decomposing the $\Gamma$, $\Delta$, and $\Upsilon$ factors, without needing a KLD penalty. As you correctly point out, this means either of the following is happening:
> > 1. $\beta$-VAE is not the best performing disentangling method and better disentangling constraints should be used instead -- e.g., works of Chen et al. (2018) and Lopez et al. (2018).
> > 2. It is theoretically impossible to achieve disentanglement without some supervision (Locatello et al, 2019) ... which might not be possible to provide in this task.
> > Exploring these options is beyond the scope of this paper and is a great future direction for this work.
> >
> > We have updated the paper to include the above discussions; see the green text on page 8 (marked as “R1 & R3 & R4”), and the Future Work in green text on page 9 (marked as “R1 & R3”), as well as the newly added Appendix A.5 in (marked as “R1 & R3”).
> >
> > REFERENCES
> > - Ricky TQ Chen, Xuechen Li, Roger B Grosse, and David K Duvenaud. Isolating sources of disentanglement in variational autoencoders. In *NeurIPS*, 2018.
> > - Francesco Locatello, Stefan Bauer, Mario Lucic, Gunnar Raetsch, Sylvain Gelly, Bernhard Schölkopf, and Olivier Bachem. Challenging common assumptions in the unsupervised learning of disentangled representations. In *ICML*, 2019.
> > - Romain Lopez, Jeffrey Regier, Michael I Jordan, and Nir Yosef. Information constraints on auto-encoding variational bayes. In *NeurIPS*, 2018.
> >
> >
> > **Explain why H-VAE-CI has a considerable gap to the other models at $\gamma=0$.**
> >
> > WIth $\gamma=0$, we get discriminative only models. Hence, we view your question as asking: *“Why is one discriminative model performing better than other discriminative models?”* … which is a fundamental question in Machine Learning in general. It is in fact quite common for models from the same family (e.g., neural network (NN)) with different architectures to perform differently (e.g., a fully connected NN vs. a convolutional NN). In case of our paper, H-VAE-CI seems to be more resilient to different $\gamma$ values in general (including $\gamma=0$), compared to {S,P}-VAE-CI -- note H-VAE-CI’s better $\textrm{PEHE}$, particularly for $\gamma \leq 0.01$.
> >
> > Viewed from another perspective, it appears that having the $L_{\textrm{VAE}}$ term is more important for {S,P}-VAE-CI than for H-VAE-CI to perform well. We hypothesize that H-VAE-CI already learns expressive representations $Z_3$ and $Z_7$, meaning the optimization no longer really *requires* the $L_{\textrm{VAE}}$ term to impose that. This is in contrast to $Z_1$ in S-VAE-CI and $Z_1$ and $Z_3$ in P-VAE-CI. To check if this hypothesis holds, we tried to calculate the mutual information (following Belghazi, et al. (2018)’s method) between $\[Z_3, Z_7\]$ and $X$ (i.e., $ MI(\ \[ Z_3, Z_7 \], X \ ) $) for H-VAE-CI, and compare the results with $ MI(\ \[ Z_1, Z_3 \], X\ ) $ for P-VAE-CI as well as $ MI(\ Z_1, X\ ) $ for S-VAE-CI. Unfortunately, however, it appears that Belghazi, et al. (2018)’s method for estimating MI does not work for high-dimensional data such as ours. We appreciate any suggestions you might have to test this hypothesis.
> >
> > We have updated the paper on pages 7 and 9 in green text (marked as “R1 & R4”), to touch on this analysis.
> >
> > REFERENCE
> > - Belghazi, M. I., Baratin, A., Rajeshwar, S., Ozair, S., Bengio, Y., Courville, A., & Hjelm, D. (2018). MINE: Mutual Information Neural Estimation. In *ICML*.
> >
> >
> > **How is the IPM chosen for $\mathcal{L}_{\mathtt{disc}}$?**
> >
> > The IPM we used in our experiments is the linear MMD. Note that we have developed our code on top of the CFR’s code-base, available online at https://github.com/clinicalml/cfrnet, from which we inherit the helper functions such as calculation of the $\mathcal{L}_{\mathtt{disc}}$. This is why our supplementary material only included the files for our proposed methods’ tensorflow graph. See the comment at the top of our published codes.

---

> > > ### Comment · AnonReviewer1 · 2020-11-20
> > > **Thanks**
> > >
> > > Thanks for the detailed clarifications, the additional experiments, and additional discussions in the paper. I agree that they have much improved the paper. I have increased my score accordingly.

---

> > > > ### Author Response · Authors · 2020-11-21
> > > > **Thank you!**
> > > >
> > > > Thank you for your great comments, and for your subsequent re-evaluation!
> > > >
> > > > Please feel free to forward any other suggestions.

---

### Official Review · AnonReviewer3 · 2020-10-23

**Rating:** 7
**Confidence:** 3

**Review:**

Summary: Some generative models have been proposed for causal effect estimation but they often do not have a competitive performance. Recent work suggested that a combination of generative and discriminative model may improve treatment estimation with observational data, and further suggests a generic latent variable model for factorizing selection bias, as well as outcome. The author(s) build on this work and propose a set of deep generative models, with a hybrid objective function (generative + discriminative), that outperforms current approaches for ATE.

The proposed family of models is appealing. The proposed objective function is motivated from previous literature. I think it is always a shame (but alright) that one must add those beta constraints on VAEs in order to get sensible results (see my question below). Still, the experimental section makes a strong point that the method is effective and outperform previous approaches.

Question:
1. What are the results of the hyperparameter search for beta? In particular, if the result of beta is small (or zero), can the model still be called a VAE, as it should degenerate as an autoencoder. In that case, the approximate posterior may have near zero variance and the model barely capture any uncertainty. I wish to understand (a) whether the resulting model is still able to really generate data, from the prior predictive density and (b) what would be the performance of the model with a fixed beta, or a beta annealing as an ablation study.
2. Would it be more advantageous to use other disentanglement constraints instead of the beta-VAE? [1, 2]

[1] https://arxiv.org/pdf/1802.04942.pdf
[2] https://arxiv.org/abs/1805.08672

---

> ### Author Response · Authors · 2020-11-18
> **We have updated the paper to address your great questions / suggestions.**
>
> Thank you for your positive opinion and great feedback. We highly appreciate it.
>
>
> **Analyse results of the ablation study for $\beta=0$; is the resulting model still able to generate data, from the prior predictive density?**
>
> This is a great point. Our initial hypothesis in using $\beta$-VAE was that it might help *further* disentangle the underlying factors. However, Figure 6(b) suggests that close-to-zero or even zero $\beta$s also work effectively. To further explore this hypothesis, we examined the decomposition tables (similar to Figure 4) of H-VAE-CI for configurations with $\beta=0$ and observed that they were all effective at decomposing the underlying factors $\Gamma$, $\Delta$, and $\Upsilon$ -- similar to the performance reported in the green table in Figure 4. The newly added Figure 9 in the Appendix shows several of these tables. Our interpretation here is that the H-VAE-CI's architecture already takes care of decomposing the $\Gamma$, $\Delta$, and $\Upsilon$ factors, without needing the help of a KLD penalty.
>
>
> **Is it more advantageous to use other disentanglement constraints instead of the beta-VAE?**
>
> In light of the above discussion, and as you correctly point out, this means either of the following is happening:
> 1. $\beta$-VAE is not the best performing disentangling method and better disentangling constraints should be used instead -- e.g., works of Chen et al. (2018) and Lopez et al. (2018); thank you for the pointers by the way.
> 2. It is theoretically impossible to achieve disentanglement without some supervision (Locatello et al, 2019) ... which might not be possible to provide in this task.
>
> Exploring these options is out of the scope of this paper and is a great future direction for this work.
> Please see the updated green text on page 8 (marked as “R1 & R3 & R4”), as well as the newly added Appendix A.5 (marked as “R1 & R3”).
>
> REFERENCES
> - Ricky TQ Chen, Xuechen Li, Roger B Grosse, and David K Duvenaud. Isolating sources of disentanglement in variational autoencoders. In *NeurIPS*, 2018.
> - Francesco Locatello, Stefan Bauer, Mario Lucic, Gunnar Raetsch, Sylvain Gelly, Bernhard Schölkopf, and Olivier Bachem. Challenging common assumptions in the unsupervised learning of disentangled representations. In *ICML*, 2019.
> - Romain Lopez, Jeffrey Regier, Michael I Jordan, and Nir Yosef. Information constraints on auto-encoding variational bayes. In *NeurIPS*, 2018.

---

> > ### Comment · AnonReviewer3 · 2020-11-23
> > **Thank you**
> >
> > I would like to thank the authors for their answers. I have kept my score.

---

> > > ### Author Response · Authors · 2020-11-24
> > > **Thank you!**
> > >
> > > Thank you for your great feedback.

---

### Official Review · AnonReviewer4 · 2020-10-25
**Proposes some interesting architectures, but novelty limited and motivations/conclusions not strongly supported by experiments**

**Rating:** 6
**Confidence:** 4

**Review:**

The paper proposes three VAE architectures for ATE estimation.

The approach is motivated by CEVAE which uses a VAE to learn a single latent representation of confounding between the treatment, target and covariates, but attempts to disentangle the confounding between the covariates and the treatment, confounding between the covariates, treatment and target, confounding between the covariates and the target and covariates-only information.

To attempt to to this, 3 different hierarchical VAE architectures are proposed which involve stacking M1 and M2 units. They also include an MMD term in the loss to encourage independence between one of the latent representations and the target.

The proposed method is evaluated using synthetic data and on two real datasets. In both cases, the proposed method outperforms the SoTA methods, but the results are only significant in one case.

The paper is generally well written. The problem and general motivation are clearly stated. However, the specific motivations for the 3 specific architectures tested could be more rigorously explained and backed up experimentally.

Given that this is more of an architecture paper, the evaluation of the specific architecture choices could be more thorough. The attempt to account for the contributions of the different latent factors is appreciated, but the results are not strongly convincing. It seems clear that Z3/Z4 captures Gamma, but it’s not clear that any of the other architecture choices make a difference. The performance results do not seem to support that they are (they do not appear to be significantly different).

It’s also not clear what to conclude regarding the hyper parameter sensitivity. It would be interesting to see whether alpha affects the ATE estimation, i.e. whether the MMD term is making a difference.

In summary, the paper proposes 3 architectures for ATE estimation. It is well written and presents reasonable ideas, but the novelty is somewhat limited (it combines existing ideas and architectures), the performance advantage over the existing SoTA is not strongly convincing and the specific architecture choices could be better motivated and evaluated experimentally.

--Post-rebuttal--

I have increased my score in response to the authors' clarifications and additional experiments and updates added to the paper.

---

> ### Author Response · Authors · 2020-11-18
> **We have updated the paper to address your great questions / suggestions.**
>
> We thank R4 for your excellent, thought-provoking feedback. Our attempt to answer your questions (especially the one on the hyperparameters’ sensitivity analyses) resulted in a modified paper that includes strong analytical discussions and new exciting findings that substantially improved our paper. For this, we are grateful!
>
>
> **Statistical significance of the results that beat the SOTA.**
>
> Our results are better than the SOTA in all three benchmarks, and this improvement is statistically significant on both IHDP and Synthetic benchmarks. The reason why it is not significant for ACIC'18 is that, while our proposed method has small variance, the *other* competing methods have large variances in their performance. This speaks to their instability on harder datasets (notice the larger magnitude of $\textrm{PEHE}$ and $\epsilon_{\textrm{ATE}}$ on the ACIC’18 benchmark compared to the other benchmarks).
>
>
> **Specific architecture choices could be more thorough and more rigorously explained.**
>
> We should have further emphasized that our champion model is H-VAE-CI (i.e., the Hybrid model). The paper describes the Series and Parallel models only to provide our line of thought in achieving the proposed design of the Hybrid model. The new manuscript provides these motivations on pages 4 and 5 -- see the green text labeled as “R4”.
>
>
> **Z3/Z4 do capture $\Gamma$, but not clear for the other factors.**
>
> Section 4.2 and Figure 4 is our attempt to report the ability of various methods to capture disentangled representations of $\Gamma$, $\Delta$, and $\Upsilon$. We emphasize that only H-VAE-CI (and not the other methods) is capable of disentangling all three factors. The green table in Figure 4 illustrates this, as $Z_1$ has the largest ratio for $\Upsilon$, and $Z_5$ has the largest ratio for $\Delta$. We agree that $Z_3$ has a more substantial ratio compared to $Z_1$ and $Z_5$; but in all fairness, this is a measure that we developed that crudely evaluates disentanglement. More research is needed to evaluate disentanglement of the underlying factors in an observational study.
>
> Note also that the green table in Figure 4 is not a single observation; we have conducted this analysis for multiple hyperparameter settings and achieved a similar distribution of ratios (see the newly added Figure 9 in the Appendix A.5). Finally, as stated in the Future Work section, *“we are yet to invent algorithms that can learn to perfectly disentangle factors $\Delta$ and $\Upsilon$”*. Stay tuned!
>
> We have updated the last paragraph of Section 4.2 (in green text; marked as “R4”) as well as a footnote (in green text; marked as “R1 & R4”) to include more detailed discussion on this matter.
>
>
> **What to conclude regarding the hyper parameter sensitivity?**
>
> We have extensively updated the paper with detailed discussions on our ablation studies for $\alpha$, $\beta$, and $\gamma$ coefficients. Please see the newly added green texts with labels “R1 & R4” and “R1 & R3 & R4” on pages 8 and 9.

---

> > ### Comment · AnonReviewer4 · 2020-11-22
> > **Updated review**
> >
> > I appreciate the authors' additional discussions and experiments added to the paper and clarifications made in their rebuttal to address the concerns raised. I agree this improves the paper and have increased my score.

---

> > > ### Author Response · Authors · 2020-11-22
> > > **Thank you!**
> > >
> > > Thank you for your great comments, and for your subsequent re-evaluation!
> > >
> > > Please feel free to forward any other suggestions.

---

### Official Review · AnonReviewer2 · 2020-10-28
**Good work on causal inference for observational studies but a bit incremental**

**Rating:** 7
**Confidence:** 2

**Review:**

Summary:

This paper introduces a new VAE architecture for performing
causal inference. It shows superior precision on estimating
the heterogeneous effect and lower bias in estimating the
treatment effect.

Clarity:

The paper was fairly clearly written and straightforward to follow.
I think a small amount of explanation of Kingma's M1 and M2 models
would have be improved the flow as your model does build on these
semi-supervised architectures.

Technical Quality:

The method and experiments are well-thought out and very clearly
evaluate the work.

What's not clear to me is how much the extra performance is from just
having a larger model and more parameters that can be fit. The
appendix says there are 3 layers of 200 variables each but the hybrid
architecture clearly has more parameters. Are those distributed
appropriately? It's also unclear how much the other architectures were
designed to estimate the heterogeneous effect.

Significance and Originality:

The work does feel more incremental than anything but it is an
important problem and there is value in estimate treatment
effect with higher accuracy.


Minor notes:

* First sentence of Section 3. "follows a(n unknown)" -> "follows an unknown"

* Some of the figures don't look like they are vector graphics

* The code doesn't seem intended to be executed. Hopefully if a full implementation can be made available later

---

> ### Author Response · Authors · 2020-11-18
> **We have updated the paper to address your great questions / suggestions.**
>
> Thank you for your helpful feedback and suggestions. We highly appreciate it.
>
>
> **Explanation of Kingma's M1 and M2 models.**
>
> We have added a paragraph that briefly describes the M1 and M2 models in the Related Works section. Please see the teal text on page 3, marked as “R2”. Appendix A.1 presents a more detailed overview of these two models.
>
>
> **How much the extra performance is from just having a larger model?**
>
> Great question! We ran a version of CFR with deeper as well as wider networks, such that it has the exact same number of neurons as H-VAE-CI. For the Synthetic benchmark, the $\textrm{PEHE}$ improved only slightly, from $0.41 {\scriptsize(0.10)}$ to $0.39 {\scriptsize (0.08)}$, which is not statistically significant (based on Welch's unpaired t-test with $\alpha=0.05$); but the $\epsilon_{\textrm{ATE}}$ degraded from $0.027 {\scriptsize (0.020)}$ to $0.040 {\scriptsize (0.028)}$. For the IHDP and ACIC’18 benchmarks, the performance significantly degraded with bigger networks. This suggests a tradeoff between performance and complexity, and that the architecture and objective function of H-VAE-CI is indeed adding value compared to those of CFR. We have updated the performance results in Table 3 as well as Figure 5, and added a footnote in Appendix A.4 -- see the added text in orange and marked as “R1 & R2”.
>
>
> **Incremental contribution.**
>
> Stand-alone pioneering research is extremely rare -- essentially every novel research stands on the shoulders of the previous works.
>
> To further demonstrate the merit of this work and the new possibilities that it provides, we explored one of our future-work items in this rebuttal period. This framework allowed us to quickly obtain some preliminary results on incorporating a context-aware re-weighting scheme that requires the disentangled learned representation of $\Delta$ in order to compute its weights. We are happy to report that these new results beat our previously presented SOTA. See pages 6 and 8 (in purple), as well as the updated Tables 1, 2, and 3, and Figures 5 and 6, in the new version of the paper. These results suggest that our contributions are effective and useful! Thanks!
>
>
> **The code is not executable.**
>
> We have adopted the code-base of CFR, online available at https://github.com/clinicalml/cfrnet, and developed our code (the tensorflow graph of the proposed methods; provided in the supplementary material) on top of that -- see the comment on top of our published code. We have, however, modified several of their helper functions as well. But of course, in order to publish the code independently, we need to ask for the Sontag Lab’s permission; which we will, upon acceptance of the paper.

---

> > ### Comment · AnonReviewer2 · 2020-11-25
> > **Rebuttal addresses all my concerns**
> >
> > Thanks! Your response addresses enough of my concerns about this paper!

---

### Decision · Program_Chairs · 2021-01-07
**Final Decision**

**Decision:**

Reject

**Comment:**

The authors suggest a VAE model for causal inference. The approach is motivated by CEVAE (Louizos et al., 2017) which uses a VAE to learn a latent representation of confounding between the treatment, target, and covariates. This paper goes beyond this approach and tries to design generative model architectures that encourage learning disentangled representations between different underlying factors of variation inspired by Hassanpour & Greiner (2019).

The reviewers agreed that the topic will be of interest to a large group of readers. While the first version of the papers raised questions about the experimental design, several questions on the architecture design were addressed during the rebuttal period (e.g., deeper architectures). Other improvements were suggested and not adopted (e.g., alternative methods to achieve better disentanglement). The ablation studies seem to suggest that some of the loss terms are not actually needed and that non-probabilistic autoencoders (beta=0) also work well. We recommend aiming at improving the writing quality and coverage of more background material on the proposed architectures and causal factors.